# Evaluating the Relative Importance of Northern African Mineral Dust Sources Using Remote Sensing

Natalie L. Bakker[1], Nick A. Drake[1], Charlie S. Bristow[2]

[1] Department of Geography, King's College London, University of London, London, UK.
[2] Department of Earth and Planetary Sciences, Birkbeck College, University of London, London, UK

*Correspondence to*: Natalie L. Bakker (Natalie.Bakker@kcl.ac.uk)

**Abstract.** Northern African mineral dust provides the Amazon Basin with essential nutrients during boreal winter months, when the trajectory of the Saharan dust plume is towards South America. This process, however, is still poorly understood.

There is little understanding where the dust is coming from, and thus what the concentration of nutrients in the dust is. This information is vital to assess the impact it will have on the Amazon. In order to further our understanding of the problem, this study analyses northern African dust sources of the boreal winter dust seasons between the years 2015-2017. It utilises high spatio-temporal resolution remote sensing data from SEVIRI, MODIS, VIIRS, and Sentinel-2 to identify dust sources, classify them according to a geomorphic dust source scheme, and quantify the relative importance of source regions by calculating the

total dust mass they produce. Results indicate that paleolakes emit the most dust, with the Bodélé Depression as the single largest dust source region. However, alluvial deposits also produce a substantial amount of dust. During the boreal winter dust seasons of 2015-2017, ~36% of the total dust mass emitted from northern Africa was associated with alluvial deposits, yet this geomorphic category has been relatively understudied to date. Furthermore, sand deposits were found to produce relatively little dust, in contrast to the results of other recent studies.

**1 Introduction**

Mineral dust is an important component of the earth system, affecting radiative forcing, cloud properties, and playing a key role in terrestrial, oceanic, atmospheric, and biogeochemical exchanges (Harrison et al., 2001; Jickells et al., 2005; Mahowald et al., 2010). The Sahara and the Sahel are the world's largest dust source regions, emitting several hundred teragrams of mineral dust yearly (Ridley et al., 2012). Most of this dust is carried over large distances and transported west and southwest

across the Atlantic Ocean (Chiapello et al., 1997). During boreal summertime dust is blown by the trade winds from northern Africa towards the Caribbean, while during winter it takes a more southerly route towards the Amazon Basin. It has been suggested that this dust has a significant fertilising effect on the nutrient deficient Amazonian rainforest soils (Swap et al., 1992; Yu et al., 2015). In order to better understand past, current, and future impacts of North African mineral dust on the Amazon Basin, the biogeochemical properties of the dust need to be determined. Recently it has been shown that these

properties differ significantly based on the geology and geomorphology of the terrestrial source (Gross et al., 2015, 2016), and it is therefore critical to identify dust sources and their geomorphic nature precisely.

Previous dust source identification studies have used geochemistry and mineralogy of dust samples, satellite remote sensing techniques, or back trajectory analysis (Schepanski et al., 2012; Scheuvens et al., 2013; Muhs et al., 2014). In the latter case, performance of trajectory models is highly dependent on the initial conditions, and it is restricted by the quality of the input data (Gebhart et al., 2005). Tracing sources using the geochemistry and mineralogy of dust samples proves difficult for Saharan dust, as samples are sporadic in time and space, and transport affects measurements in poorly understood ways (Formenti et al., 2011). Satellite remote sensing on the other hand, provides high spatio-temporal data across the region, and has been found to be an effective method of identifying dust source areas (Prospero et al., 2002; Washington et al., 2003; Schepanski et al., 2012). Notwithstanding this, many remote sensing techniques identify erroneous sources as they average both emission and transport (Prospero et al., 2002; Washington et al., 2003), and thus methods need to be adopted which only identify the site of dust emission (Schepanski et al., 2012; Ashpole & Washington, 2013).

While the major dust source areas within the Sahara and the Sahel have been identified using remote sensing (Prospero et al., 2002; Schepanski et al., 2009; Ginoux et al., 2012; Ashpole & Washington, 2013), only a few studies classify the geomorphology of these sources (Crouvi et al., 2012; Ashpole & Washington, 2013), and knowledge of the relative importance of the major dust regions and is lacking. Studies analysing the fertilisation effect of dust on the Amazon focus primarily on the impact of paleolake Bodélé (Bristow et al., 2010; Yu et al., 2015) as this has been shown to be the most important source region (Koren et al., 2006; Ben-Ami et al., 2010). Latest estimates of phosphorous input into the Amazon are based solely on the phosphorous concentration in Bodélé dust (Yu et al., 2015), but this is unlikely to reflect the diversity of the surfaces producing dust.

The aim of this study is to identify wintertime dust sources, as only this dust is transported to the Amazon. The sources are classified according to the geomorphic dust source scheme developed by Bullard et al. (2011), after which the importance of the various dust sources is quantified, and it is calculated how much dust each region produces. This is achieved through a combination of remote sensing products with different temporal and spatial resolution, to optimise the identification of dust storms, dusts source regions and the geomorphology of individual dusts sources. The Spinning Enhanced Visible and Infrared Imager (SEVIRI) dust red, green, blue (RGB) colour composite has a high temporal resolution and is used here to detect dust storms, while the Visible Infrared Imaging Radiometer Suite (VIIRS) dust RGB with higher spatial resolution is utilised to accurately determine the dust source locations. The Moderate-resolution imaging spectroradiometer (MODIS) aerosol optical depth product is employed to quantify how much dust the sources produce, and the high spatial resolution (10 m) Sentinel-2 true colour composite is used to identify the geomorphology of each source. The study examines the dust storms of the 2015-

2017 winter dust seasons. Both the number of individual dust sources of each geomorphological class, as well as the total dust mass emitted per class and dust source region are estimated.

## 2 Methodology

The study analyses northern African dust sources during boreal wintertime, defined as the area between the longitudes 20°W – 40°E and the latitudes 5°N – 40°N. The dust seasons analysed run between Dec 1st 2015 – Feb 29th 2016 and Dec 1st 2016 – Feb 28th 2017, following the wintertime dust months of Schepanski et al. (2009). Figure 1 provides a complete overview of the data and methodology.

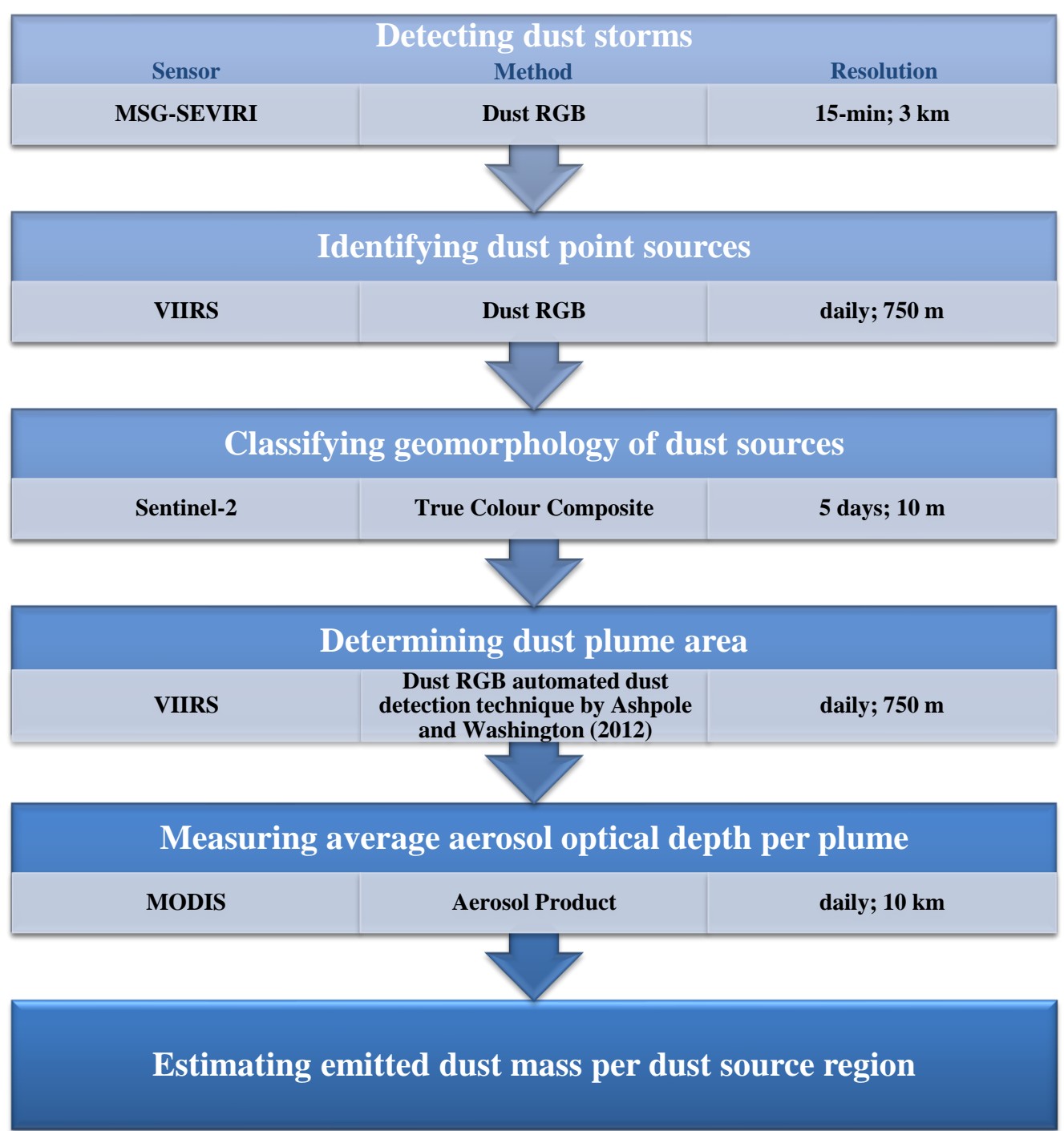

**Figure 1: Overview of the methodology of the study. The remote sensing sensor used in each step is presented in the left column, the image processing method employed is shown in the middle column, and the resolution (temporal; spatial) is shown in the right column.**

## 2.1 Detecting dust storms

MSG-SEVRI was utilised to detect dust storms every 15 minutes at a spatial resolution of 3 km at nadir. The dust RGB method of Lensky and Rosenfeld (2008) was employed whereby infrared channels are utilised to produce an image in which dust is portrayed in a pink colour. To achieve this, the red band is allocated the brightness temperature difference (BTD) between
channels at 12 - 10.8 μm, green the BTD between channels at 10.8 - 8.7 μm, and blue the brightness temperature (BT) of the 10.8 μm channel. Processed SEVIRI dust RGB images were acquired from the database of the FENNEC project (FENNEC, 2017). By visually interpreting images, all dust events of the 2015-2017 boreal winter dust seasons were identified. Though the high temporal resolution of SEVIRI makes it suitable for observing the evolution of dust storms (Schepanski, 2009, 2012; Ashpole & Washington, 2012), its low spatial resolution means it is less suited to determine the precise location and nature of
the sources.

## 2.2 Identifying dust point sources

VIIRS was used to identify dust point sources as it is currently the sensor with the highest spatial resolution that has a daily temporal resolution; it has 5 imaging-resolution bands (I-bands) at 375 m resolution, and 16 moderate-resolution bands (M-bands) at 750 m resolution. The same dust RGB method was employed using the infrared M-bands, centred at 8.55 μm, 10.76
15   μm, and 12.01 μm. Dust point sources were identified manually using visual interpretation. The individual dust plumes were tracked upwind back to their first point of occurrence (see Figure 2), except for those forming under clouds (dust storms with sources under clouds were visible on approximately 2 days per month, and this was thus not a significant problem). As this process is labour intensive, this step was performed on only the ten largest dust storms (in areal coverage) of each dust season; i.e. a total of 20 dust storms were analysed in this step.

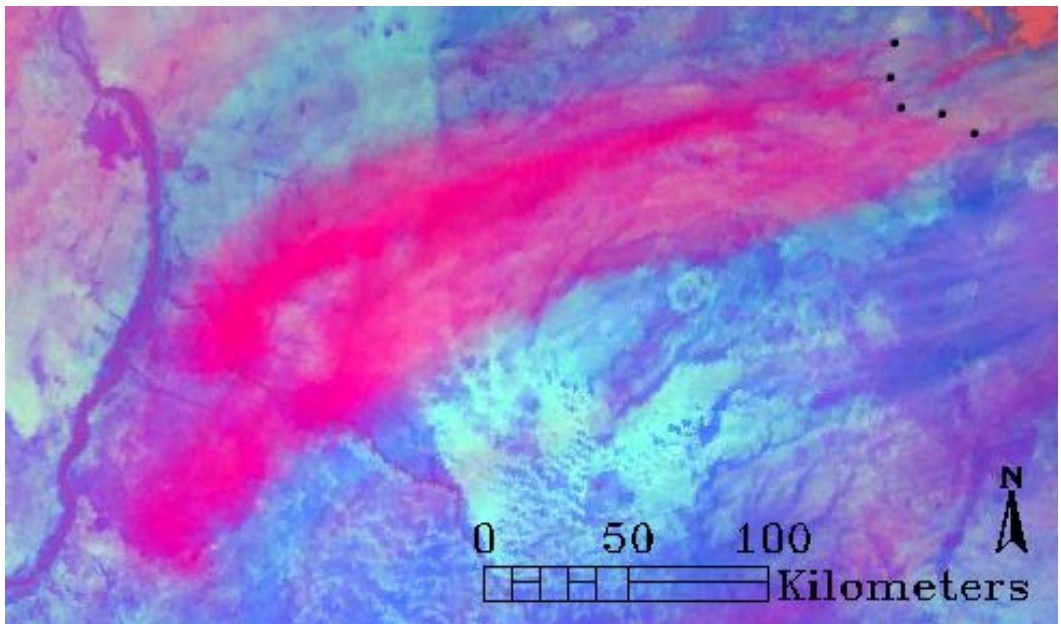

**Figure 2: VIIRS Dust RGB (Lensky & Rosenfeld, 2008) showing dust being emitted northeast of the Sudanese Nile (lat: 18, lon: 35) on December 19, 2016. Dust point source locations are identified by tracking dust plumes back to their first point of occurrence.**

### 2.3 Classifying geomorphology of dust sources

The geomorphology of the dust point sources of the 20 largest dust storms was analysed using Sentinel-2 imagery. This sensor collects visible, near infrared, and shortwave infrared imagery at a spatial resolution of 10 – 60 meters, and a temporal resolution of 5 days. Using the 10-meter resolution true colour composites, it is possible to distinguish many geomorphic features within the imagery. Cloud free, level 1c Sentinel-2 data was obtained for each of the dust point source locations identified from the VIIRS data. Each point source was interpreted and allocated to one of the geomorphic classes described

by Bullard et al. (2011). These classes are based on differences in surface characteristics governing their susceptibility of aeolian erosion, provided they met the criteria to be readily identifiable from remote sensing data. The classes from the scheme identified in this study are 'paleolakes', 'alluvial deposits', 'stony surfaces', and 'sand deposits'. Furthermore, the class 'anthropogenic' was added to the classification scheme, defined as dust sources linked to urban areas, villages, roads, and agricultural lands. Examples of the various geomorphic classes can be found in Appendix i. Drake et al. (2008)

demonstrated that the geomorphic classification scheme could characterise dust sources in the Western Sahara region successfully. To account for the spatial resolution differences between VIIRS and Sentinel-2, point sources were classified as the most prominent feature and/or most common geomorphic class within 750 m of the point location determined with VIIRS.

The identified point sources tended to spatially cluster according to their geomorphological type into 12 major dust source regions (Figure 3). In most regions, a single geomorphological source class dominated, and a predominant geomorphology per region could be determined. In two cases, namely in the South Air and El Eglab dust source regions, there was no single predominant geomorphic class, and they were thus classified as mixed paleolake and alluvial deposit (50/50%). Once the

major dust source regions were established, all dust plumes of the 2015-2017 wintertime dust seasons were assigned to the nearest major source region based on its upwind edge and then assigned the predominant geomorphic class of the region they originated from. A further four prominent dust source regions were added (see regions with dashed outlines in Figure 3), which emitted dust plumes during the two winter dust seasons, although they did not emit during the 20 largest dust storms. The predominant geomorphology of these regions was determined separately through visual interpretation of Sentinel-2 data

using the same methods that were employed for the 20 largest dust storms.

The accuracy of using only the 20 largest dust storm days of the two seasons to determine the major dust regions and their predominant geomorphology was evaluated by randomly selecting three additional days (29 Dec 2015, 21 Jan 2017, 8 Feb 2017), identifying all point sources of these days, analysing the geomorphic class of each detected point source, and

subsequently determining what percentage of these dust point sources coincided with that defined by the map of predominant dust geomorphologies. The geomorphological class of a further 163 dust point sources were identified from the three randomly chosen days. Of the 163 dust point sources, 161 (98.77%) corresponded with the predominant geomorphology assigned to the region.

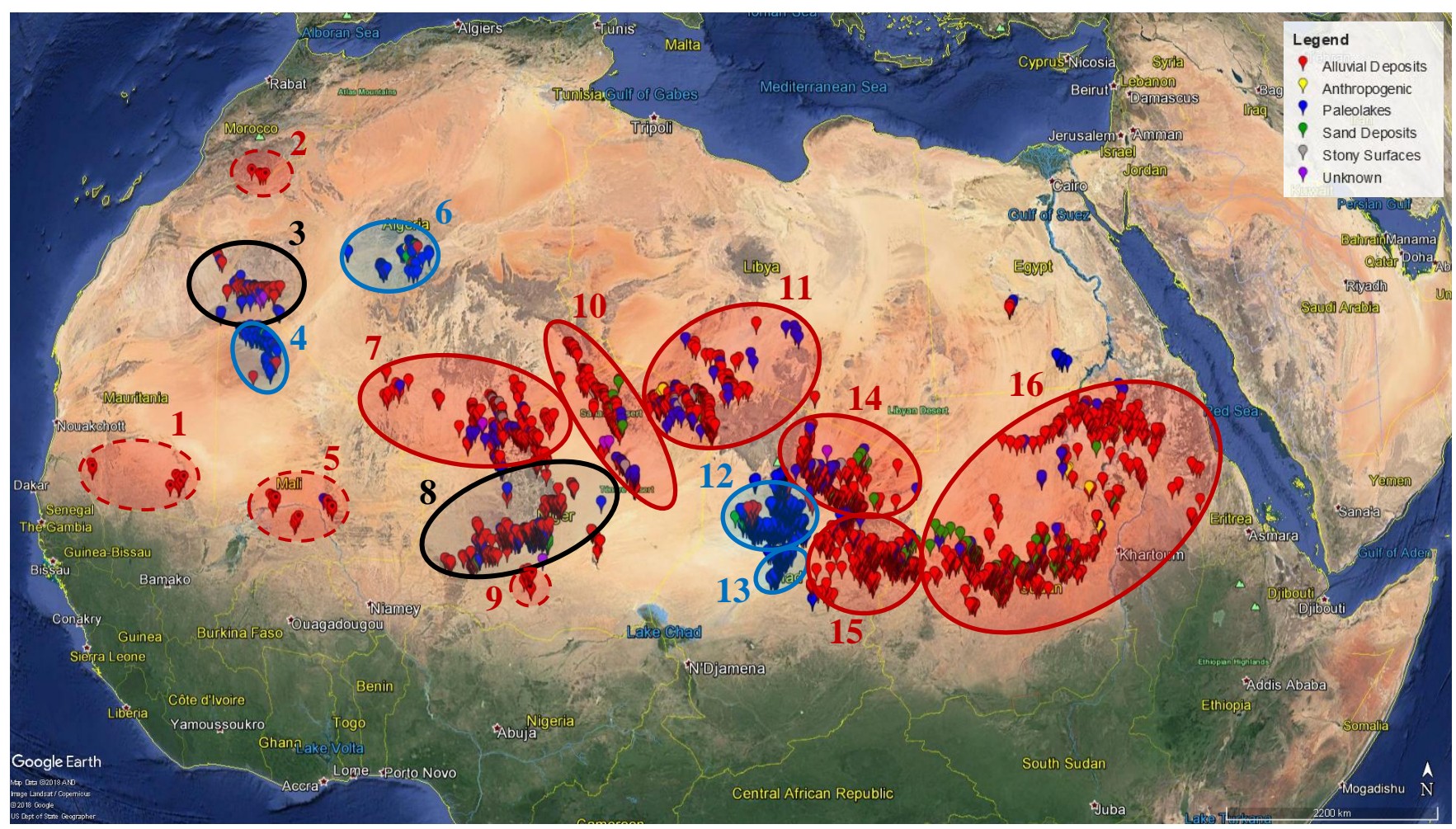

**Figure 3. Geomorphology of the northern African mineral dust sources of the 2015-2017 boreal winter dust seasons. Coloured pins represent individual dust point sources of the 20 largest dust storms of the 2015-2017 winter dust seasons, with the colour signifying the geomorphic category of the dust point source. The major dust source regions are outlined, with red regions containing predominantly alluvial deposit dust sources, blue regions containing predominantly paleolake dust sources, and black regions containing a mixture of alluvial deposit and paleolake dust sources. The regions with dashed outlines were regions that emitted dust during the dust seasons, but did not emit during the 20 largest dust storms of the 2015-2017 winter dust seasons. Source areas are numbered as follows: 1, Mauritania; 2, Draa River; 3, El Eglab; 4, Northwest Mali; 5, Mid Mali; 6, Algeria; 7, North Air; 8, South Air; 9, South Niger; 10, Djado Plateau; 11, Tibesti; 12, Bodélé; 13, Bahr el Gazel; 14, Upper Bodélé; 15, Ennedi; 16, Sudan. Background Google Earth, imagery Landsat/Copernicus (2018).**

## 2.4 Determining dust plume area

After the main dust source regions were determined, the next step was to estimate how much dust was emitted per region. To do so, firstly, the dust plume area was calculated for all dust plumes of the 2015-2017 wintertime dust seasons using the automated dust detection technique developed by Ashpole and Washington (2012). In this method, the following thresholds are applied to the brightness temperatures (in Kelvin) of the 10.76, 12.01, and 8.55 μm channels of VIIRS imagery:

$$BT\ 10.76 \geq 285$$
$$BTD\ (12.01\text{-}10.76) \geq 0 \tag{1}$$
$$BTD\ (10.76\text{-}8.55) \leq 10$$

Background surface features that were incorrectly detected as dust were removed manually, after which dust plumes could be outlined and their size calculated. The methodology is objective, easily reproducible, fast, and was found to be consistent for moderate-to-heavy dust storms occurring during daytime hours (Ashpole & Washington, 2012). The method was applied to VIIRS, rather than SEVIRI or MODIS, due to its higher spatial resolution and its seamless continuation between swaths.

## 2.5 Measuring average aerosol optical depth per plume

In order to determine the concentration of dust within the dust plumes, average (arithmetic mean) aerosol optical depth (AOD) per dust plume was calculated for all dust storms of the 2015-2017 wintertime seasons. As the VIIRS AOD product was found to contain large data gaps, particularly over bright desert surfaces, the MODIS aerosol product (MOD04_L2, MYD04_L2) was employed. MODIS monitors AOD on a daily basis at a 10-km spatial resolution, and combines the MODIS Dark Target (Kaufman et al., 1997; Levy et al., 2013) and MODIS Deep Blue (Hsu et al., 2004) algorithms. MODIS Dark Target uses the spectral radiances between 0.47 and 2.1 μm to measure AOD over oceans and dark land surfaces. MODIS Deep Blue is designed to analyse aerosols over bright-reflecting surfaces, such as deserts, utilising measurements from the blue wavelengths (412 and 470 nm). The combined algorithms allow MODIS to record data across North Africa consistently (Sayer et al., 2013).

The methodology presented here does have a few limitations. These are: 1) The once daily temporal resolution of MODIS, as dust storms in northern Africa can form during any time of the day (Schepanski et al., 2012), and this thus causes a few dust storms to be missed. Nevertheless, the vast majority of dust storms in northern Africa initiate in the morning between the hours of 06.00-12.00 (Schepanski et al., 2009), and the dust storms in this study were observed to span several hours, such that the MODIS mean overpass time of 13.30 local time did cover nearly all dust emissions; albeit dust plumes emitting after 13.30 were missed (which occurred on 1-2 days per season). 2) The MODIS and VIIRS overpass times are close but not identical, varying up to 20 minutes. 3) Dust storm days between 19-27 February 2016 had to be omitted from this study due to a MODIS technical outage, during which the satellite did not log any data, and there was hence no MODIS AOD data available.

**2.6 Estimating emitted dust mass per dust source region**

The dust mass for all dust plumes of the 2015-2017 wintertime seasons was estimated by combining the dust plume area data with the average MODIS AOD. The dust column concentration equation by Kaufman et al. (2005) was applied to all dust storms, whereby dust mass is determined using the relationship between plume size and average dust AOD at 550 μm:

$$M_{du} = 2.7A\tau_{du} \tag{2}$$

where $M_{du}$ is dust mass, A is the dust plume area, and $\tau_{du}$ is the average AOD at 550 μm. The coefficient of 2.7 is derived through the extinction efficiency of 0.37 measured by Haywood et al. (2003). However, a more recent review of mass extinction efficiency measurements by Ansmann et al. (2012) shows higher reported values ranging between 0.37- 0.57. An updated Kaufman equation was derived using the extinction efficiency of 0.57, resulting in a coefficient of 1.75:

$$M_{du} = 1.75A\tau_{du} \tag{3}$$

Using equations 2 and 3, the dust mass range was estimated (i.e. eq. 3 for the lower limit, eq. 2 for the upper limit). Ben-Ami et al. (2010) demonstrated that the Kaufman equation could successfully be used to calculate dust mass over northern Africa, although it should be noted that the equation has a relatively high uncertainty of ±30% for AOD values between 0.2 and 0.4 and has potentially even greater errors for AOD values above 0.4.

Once the dust mass per dust plume was estimated, dust plumes were grouped per dust source region and per geomorphology, and as such the total dust mass per region and dust mass per geomorphic class could be estimated.

**3 Results**

From the 20 largest dust storms, a total of 3512 individual dust point sources were identified (see Figure 3). The dust point sources per geomorphological class are shown in Table 1; the vast majority were associated to alluvial deposits (51.11%) or 20 paleolakes (42.85%). Sand deposits, including dune fields and sand sheets, accounted for only 3.93%. A few sources were linked to anthropogenic influences, such as villages, dirt roads, or agricultural fields. Moreover, 22 point sources had no clear distinguishable features to place them in one of the geomorphic categories, and were thus labelled as 'unknown'.

**Table 1. Northern African mineral dust point sources of the 2015-2017 winter dust seasons classified by geomorphology type.**

| DUST POINT SOURCE GEOMORPHOLOGY TYPE | COUNT 15-16 | % | COUNT 16-17 | % | TOTAL COUNT | % OF TOTAL |
|---|---|---|---|---|---|---|
| Alluvial deposits | 1293 | 53.47 | 502 | 45.89 | 1795 | 51.11 |
| Paleolakes | 992 | 41.03 | 513 | 46.89 | 1505 | 42.85 |
| Sand deposits | 82 | 3.39 | 56 | 5.12 | 138 | 3.93 |
| Stony surfaces | 19 | 0.79 | 15 | 1.37 | 34 | 0.97 |
| Unknown | 18 | 0.74 | 4 | 0.37 | 22 | 0.63 |
| Anthropogenic | 14 | 0.58 | 4 | 0.37 | 18 | 0.51 |
| TOTAL | 2418 | | 1094 | | 3512 | |

During the 2015-2017 winter dust seasons a total of between 82.3 - 127.0 Tg of dust was emitted (see Table 2). During the 2015-2016 dust season 54.6-84.3 Tg of dust was emitted, of which the ten largest dust storms contributed 33.3%, while the ten largest dust storms of the 2016-2017 dust season produced 45.8% of the 27.7-42.7 Tg emitted that season. The 2015-2016 dust season was significantly more dusty, producing approximately twice the amount of dust compared to the 2016-2017 dust season, which is also reflected in the amount of individual dust point sources identified per season in Table 1. The distribution of the dust over the various regions was, however, relatively similar in both seasons.

Table 2 shows the distribution of the emitted dust per dust source region, along with the frequency of emission, i.e. the percentage of days during which the region was an active dust source, the mean AOD, and the predominant geomorphology. The Bodélé Depression is identified as the single largest source region, emitting 44.5% of the dust mass, and producing the densest dust plumes with an average (arithmetic mean) AOD value of 0.91. The Bodélé, Bahr el Gazel, and Ennedi regions were the most frequently emitting dust source regions, all of them emitting during more than half of the days of the wintertime dust season. Mauritania and South Niger were the least active regions, emitting dust on only 2 of the 172 days analysed.

Dust source regions of predominantly paleolake origin emitted approximately 64% of the total dust mass during the 2015-2017 winter dust seasons, compared to 36% for the dust regions governed by alluvial influence (note that South Air and El Eglab are counted as 50% paleolake and 50% alluvial deposit). The importance of paleolakes and alluvial deposits, however, fluctuated considerably. Figure 4 compares the influence of paleolakes versus alluvial deposits for the ten largest dust storms of the 2016-2017 winter dust season, which demonstrates this variability.

**Table 2. Total dust mass emitted per Saharan dust source region during the 2015-2017 winter dust seasons, alongside their predominant geomorphology, mean AOD, and their frequency of emission.**

| | TG 15-16 | TG 16-17 | TG TOTAL | % OF TOTAL | FREQUENCY (% DAYS ACTIVE) | MEAN AOD | PREDOMINANT GEOMORPHOLOGY |
|---|---|---|---|---|---|---|---|
| **Bodélé** | 24.7 - 38.1 | 11.9 - 18.4 | 36.6 - 56.5 | 44.5 | 67.4 | 0.91 | Paleolake |
| **Bahr el Gazel** | 5.9 - 9.2 | 2.8-4.3 | 8.7 - 13.5 | 10.6 | 55.2 | 0.74 | Paleolake |
| **South Air** | 5.0 - 7.8 | 3.3-5.0 | 8.3 - 12.8 | 10.1 | 43.6 | 0.40 | Alluvial Deposit/ Paleolake |
| **North Air** | 5.3-8.2 | 1.5-2.3 | 6.8 - 10.5 | 8.3 | 25.0 | 0.31 | Alluvial Deposit |
| **Sudan** | 4.6-7.0 | 1.6-2.5 | 6.2 - 9.5 | 7.5 | 36.6 | 0.19 | Alluvial Deposit |
| **Ennedi** | 2.7-4.2 | 2.1-3.2 | 4.8 - 7.4 | 5.8 | 54.7 | 0.28 | Alluvial Deposit |
| **Algeria** | 1.3-2.0 | 1.1-1.7 | 2.4 - 3.7 | 2.9 | 12.8 | 0.32 | Paleolake |
| **Tibesti** | 1.8-2.7 | 0.6-1.0 | 2.4 - 3.7 | 2.9 | 24.4 | 0.25 | Alluvial Deposit |
| **Djado Plateau** | 1.1-1.7 | 0.4-0.7 | 1.5 - 2.4 | 1.9 | 12.2 | 0.37 | Alluvial Deposit |
| **El Eglab** | 0.4-0.6 | 0.7-1.0 | 1.1 - 1.6 | 1.3 | 5.2 | 0.35 | Alluvial Deposit/ Paleolake |
| **Mid Mali** | 0.1-0.2 | 0.8-1.2 | 0.9 - 1.4 | 1.1 | 4.7 | 0.42 | Alluvial Deposit |
| **Upper Bodélé** | 0.6-0.9 | 0.2-0.4 | 0.8 - 1.3 | 1.0 | 22.7 | 0.22 | Alluvial Deposit |
| **Draa River** | 0.7-1.0 | 0 | 0.7 - 1.0 | 0.8 | 1.7 | 0.31 | Alluvial Deposit |
| **South Niger** | 0 | 0.5-0.8 | 0.5 - 0.8 | 0.6 | 1.2 | 0.65 | Alluvial Deposit |
| **Mauritania** | 0.3-0.5 | 0.1-0.1 | 0.4 - 0.6 | 0.5 | 1.2 | 0.54 | Alluvial Deposit |
| **Northwest Mali** | 0.1-0.2 | 0.1-0.1 | 0.2 - 0.3 | 0.2 | 2.3 | 0.36 | Paleolake |
| **TOTAL** | 54.6-84.3 | 27.7-42.7 | **82.3 - 127.0** | | | | |

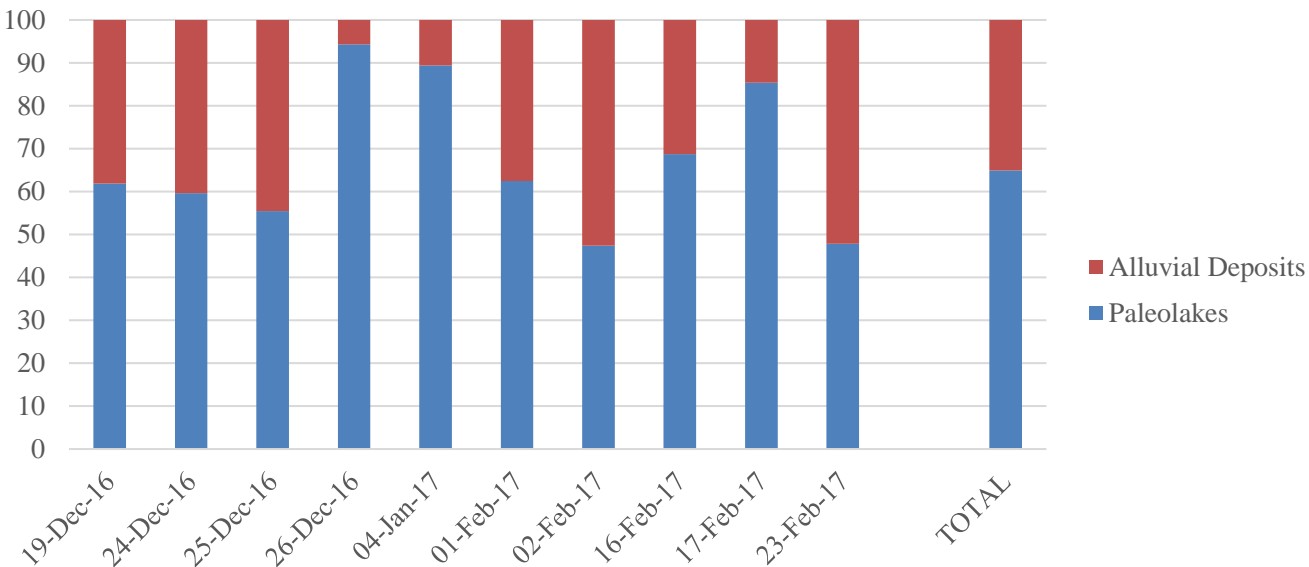

**Figure 4. Contribution (in %) of paleolake dust sources versus alluvial deposit dust sources to total dust mass for the ten largest dust storm events of the 2016-2017 winter dust season.**

## 4 Discussion

### 4.1 Paleolake and alluvial deposit dust sources

Our results indicate the importance of both paleolakes and alluvial deposits as African mineral dust sources. Alluvial deposits were found to deflate more often, while paleolakes produced more dust by emitting denser dust plumes. Dry lakebeds, most notably the Bodélé Depression, are widely known to be preferential dust sources (Prospero et al., 2002; Tegen et al., 2002). Alluvial deposits, however, have been recognised to a much lesser extent in dust source studies, especially when compared to paleolakes (Bullard et al., 2011). Producing 51.11% of the individual dust plumes and representing ~36% of the total dust mass, alluvial deposits make a significant contribution to the dust burden. Despite this, the latest estimates of the fertilisation effect of dust on the Amazon by Yu et al. (2015) determine total phosphorus deposition into the Amazon (0.022 Tg per annum) based solely on the phosphorous concentration in Bodélé dust, while not taking alluvial deposit dust sources into account. Given the fact that it has recently been shown that there are significant differences of bioavailable phosphorous content within dust from different source geomorphologies (Gross et al., 2015; 2016), and the evidence presented here of alluvial deposits being important sources, the accuracy of Yu et al.'s (2015) estimate can be questioned. In Namibia, Dansie et al. (2017) established that alluvial dust source sediments contained up to 43 times greater concentrations of bioavailable iron and enriched nitrogen (N) and phosphorus (P) macronutrients compared to paleolake pan sediments. Further research into the actual differences in nutrient content of the different northern African dust sources and dust source geomorphologies is needed to improve the precision of fertilisation estimates. This is particularly the case for the underexamined alluvial source regions.

## 4.2 Sand dunes producing dust?

While paleolakes and alluvial deposits accounted for over 90% of the emitted dust, sand deposits only accounted for 3.93%. This contradicts recent results by Crouvi et al. (2012), who found that active sand dunes were the most frequent dust sources within the Sahara. The discrepancy between the results presented here and those found by Crouvi et al. (2012) is most likely

explained by the type of data used and the difference in their spatial resolution. Crouvi et al. (2012) mapped dust sources onto a coarse resolution (1° x 1°) grid, and compared them to a map of soils. When more than one soil type was present in a grid cell, Crouvi et al. (2012) assigned it to the geomorphic unit with the highest areal coverage. This carries a large bias towards widespread sand sheets and dune fields, whereas for instance paleorivers and small to medium-sized paleolakes (which could be picked up with the high-resolution Sentinel-2 data utilised in this study) are underrepresented and easily overlooked.

Another major difference between the studies is that Crouvi et al. (2012) classify the Bodélé region as sand dunes, whereas in this study it is classified as a paleolake. While dunes exist in the Bodélé Depression, they are commonly composed of fragments of lake sediments and the dunes themselves are subordinate in cover to lake sediment exposures (Bristow et al., 2009). Furthermore, Crouvi et al. (2012) identified quartz-rich sand dune fields surrounding the Bodélé Depression as a major source of dust, however closer examination of the dust sources in this region, using high resolution data from VIIRS and Sentinel-2,

showed this was not the case. The higher resolution data used in this study has improved the accuracy of dust source identification, and suggests that sand dunes produce little dust in northern Africa during wintertime.

## 4.3 Location of dust sources

The majority of the locations of dust sources identified in this study correspond with previous studies such as Prospero et al.

(2002), Schepanski et al. (2007; 2009; 2012), Ginoux et al. (2012), and Knippertz and Todd (2012). However, one area that has not been widely recognised as a major source region is central/southern Sudan; only Schepanski et al. (2009) mentions this region. In southern Sudan, a few of the sources lie within the Sahel, but most are located in the Sahara. While various studies have analysed Sahelian dust emissions (Klose et al., 2010; Bergametti et al., 2017; Kim et al., 2017), less is known about the importance of the Sahel compared to the Sahara. From our data, we find that the Sahelian dust sources account for a

mere 5%, with the vast majority of wintertime dust sources being located in the Sahara. Of the Sahelian dust sources, 87% are alluvial. The lack of lacustrine dust sources in the Sahel may be due to higher rainfall leading to moisture accumulation in the lake basins, which in turn suppresses dust production. The inundation of ephemeral lakes has previously been observed to lower dust emissions in semi-arid regions (Mahowald et al., 2006).

## 5 Conclusion

This study provides a quantitative analysis of wintertime northern African mineral dust sources. It outlines a novel methodology in which high temporal resolution remote sensing data is combined with high spatial resolution data to analyse

dust sources more accurately than previous dust source studies. The results show for the first time the strong importance of alluvial deposits, while simultaneously reaffirming the importance of paleolake sources. Sand deposits, on the other hand, are found to produce relatively little dust. The results concur with previous studies that the Bodélé Depression is the single largest dust source region, yet also highlight that there are lesser studied regions, such as central Sudan, producing a substantial amount of dust. It furthermore stresses the significance of the Sahara over the Sahel regarding dust production, with 95% of dust sources emitting from the Sahara.

*Data availability*

SEVIRI data available from the FENNEC project ([http://www.fennec.imperial.ac.uk/](http://www.fennec.imperial.ac.uk/), last access September 2018). VIIRS data available from NOAA CLASS ([https://www.avl.class.noaa.gov/](https://www.avl.class.noaa.gov/), last access September 2018). Sentinel-2 data can be accessed through the ESA Copernicus Open Access Hub ([https://scihub.copernicus.eu/dhus/#/home](https://scihub.copernicus.eu/dhus/#/home), last access July 2018). MODIS data is available from NASA Earthdata LAADS DAAC ([https://ladsweb.modaps.eosdis.nasa.gov/](https://ladsweb.modaps.eosdis.nasa.gov/), last access December 2018).

*Author contributions*

The conceptualisation and methodology of this research was established by N.L. Bakker and N.A. Drake. Investigation and formal analysis were carried out by N.L. Bakker, supervised by N.A. Drake and C.S. Bristow. Interpretation of results determined from discussions with all three authors. The original draft of the manuscript was written by N.L. Bakker, then reviewed and edited by N.A. Drake and C.S. Bristow.

*Competing interests*

The authors declare that they have no conflict of interest.

*Acknowledgements*

This research was carried out under Natural Environment Research Council grant NE/L002485/1. The authors would like to thank Dr. Ian Ashpole and anonymous referees for reviewing the paper and providing valuable comments and suggestions.

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

## Appendix i: Examples of geomorphic classes

In the figures below, examples of the various geomorphic classes used in this study can be found. In order, the geomorphic categories are: paleolake (Figure 5), alluvial deposit (Figure 6), stony surface (Figure 7), sand deposit (Figure 8), and anthropogenic (Figure 9).

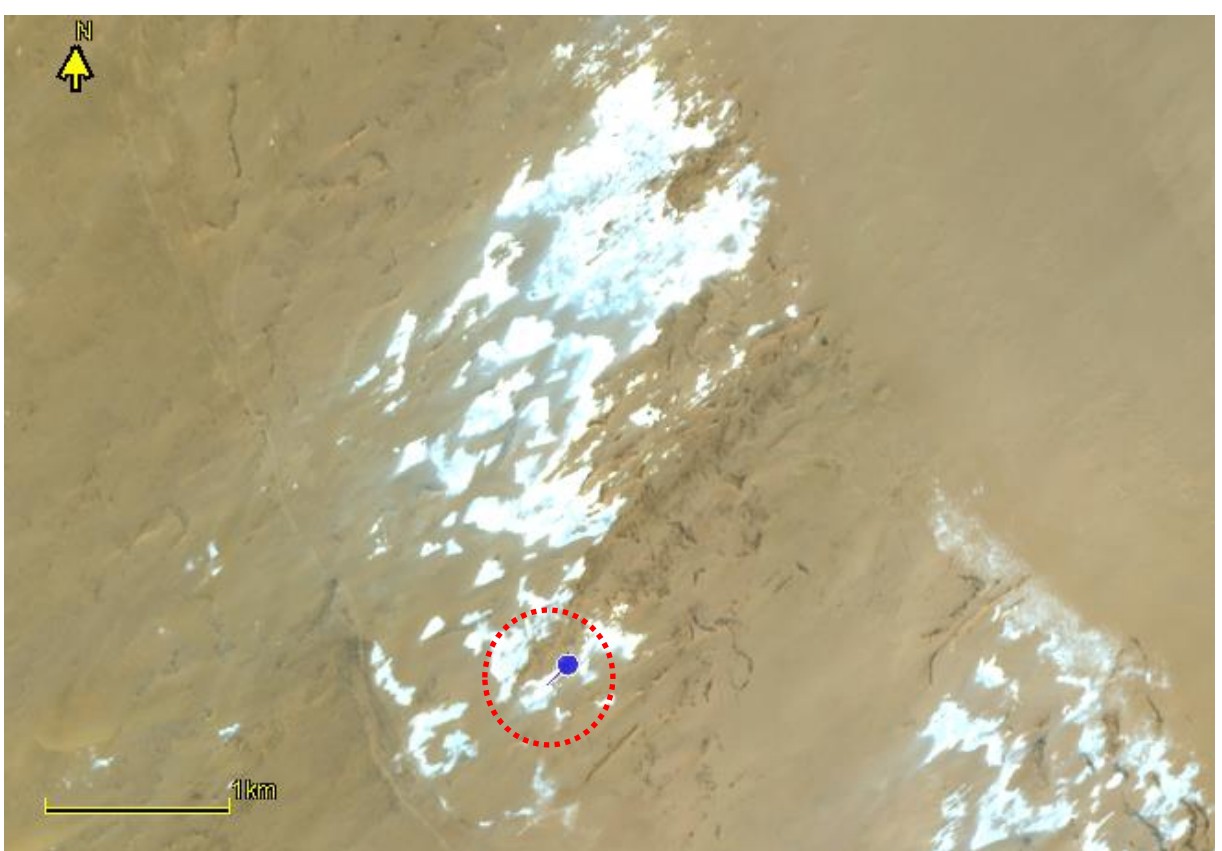

**Figure 5: Sentinel-2 natural colour composite showing a diatomite paleolake in the Bodélé Depression region, Chad, identified as a dust source. This source emitted dust on the 23rd of December 2015. Location of the pin in lat, long: 17.4875, 19.4845. 750m buffer included in red.**

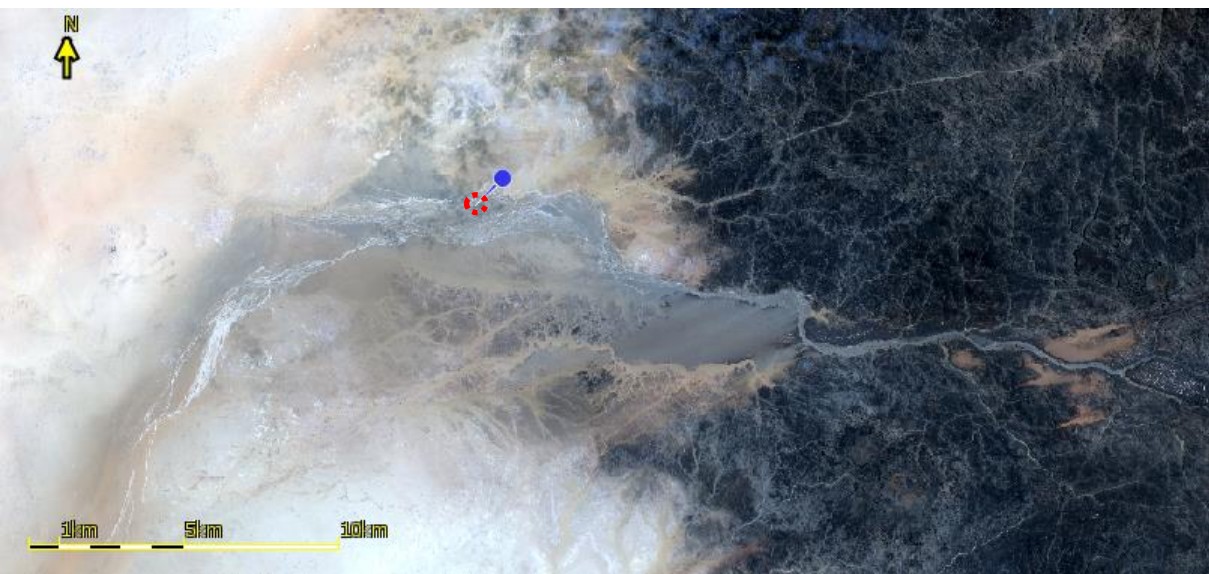

**Figure 6: Sentinel-2 natural colour composite showing an alluvial deposit dust source west of the Tibesti mountains in Chad. The source emitted dust on the 30th of November 2015. Location of the pin in lat, long: 21.4001, 15.8211. 750m buffer included in red.**

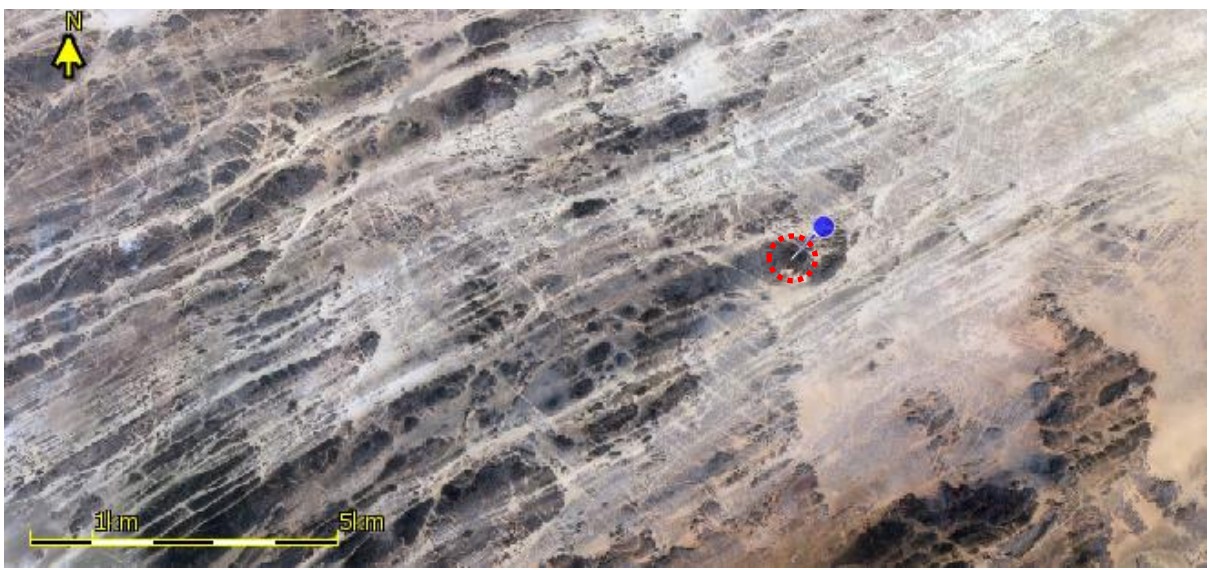

**Figure 7: Sentinel-2 natural colour composite showing a stony surface dust source near the border between Chad and Niger in the North. The source emitted dust on the 1st of December 2015. Location of the pin in lat, long: 22.2030, 15.2052. 750m buffer included in red.**

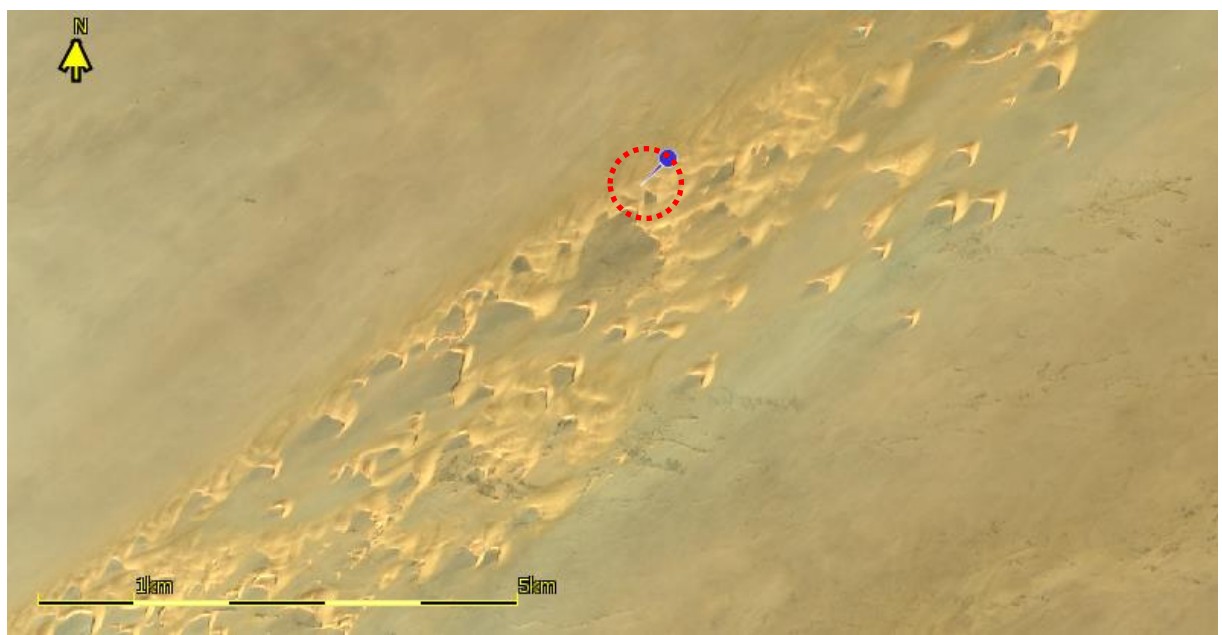

**Figure 8: Sentinel-2 natural colour composite showing a sand deposit (barchan dune) dust source, ~35 km northeast of Faya-Largeau, Chad. This source emitted dust on the 23$^{rd}$ of December 2015. Location of the pin in lat, long: 17.4655, 19.6163. 750m buffer included in red.**

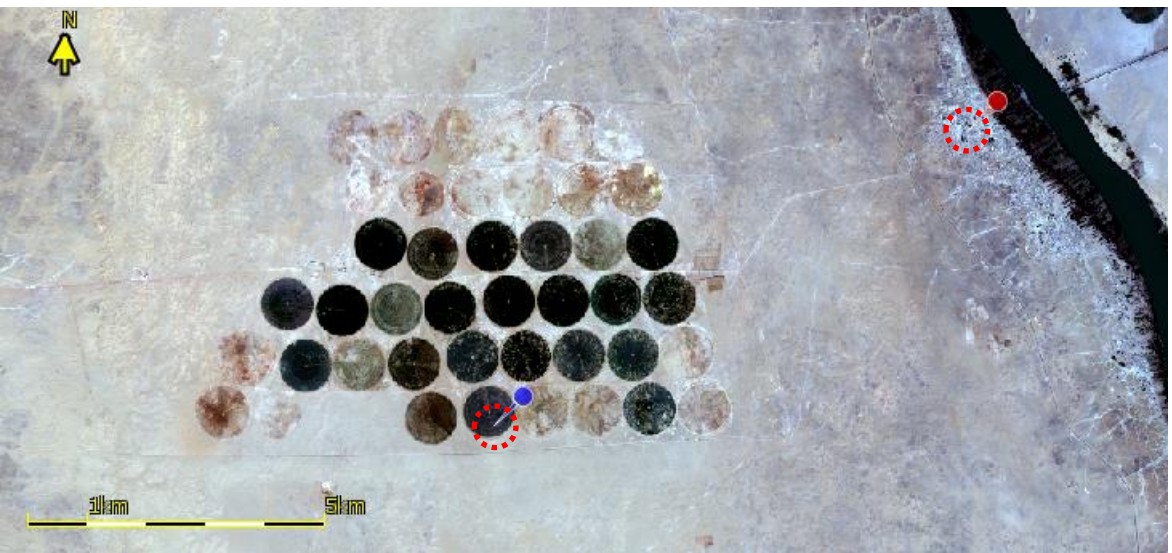

**Figure 9: Sentinel-2 natural colour composite showing two anthropogenic dust sources in Sudan. The agricultural dust source (blue pin) emitted dust on the 21st of February 2016. Location of the blue pin in lat, long: 18.7989, 30.4026. The village dust source (red pin) emitted dust on the 26$^{th}$ of January 2016. Location of the red pin in lat, long: 18.8448, 30.4776. 750m buffer included in dotted red.**

