# Peer review of "Evaluating the Relative Importance of Northern African Mineral Dust Sources Using Remote Sensing"

_Atmospheric Chemistry and Physics, 2019_

## Referee Comment (RC1) · Anonymous Referee #1 · 22 Apr 2019

The study is motivated by the question, how much phosphorus is supplied to the Amazon by North African dust sources. The authors examine the source regions for winter time dust storms (as this is the season when dust is transported towards the Amazon) for three consecutive years (2015-2017). Geomorphological characteristics of the source region were determined for the ten strongest dust storms per winter season.

The manuscript is well written and I enjoyed reading it. Nevertheless, I have a few questions I would like the authors to address before publication:

General comments:

(1) The question I am most curious to know the answer remains somewhat unan-

swered: Can you estimate how much phosphorus is supplied to the Amazon by the individual dust source regions? E.g., Schepanski et al., Atmos. Chem. Phys. (2009) examined by means of a model simulation the contribution of dust emitted from the Bodélé Depression to the atmospheric dust burden over the Gulf of Guinea and tropical Atlantic showing a contribution of up to 60% over the Equator region. But it remains unclear, how much dust passes through the tropical rain belt. I think it may be worth considering adding a trajectory analysis here showing how many dusty trajectories originating from an emitting dust source actually reach the Amazon without being affected by rainfall or clouds.

(2) Related to question (1), how many of the identified dust sources actually contribute to the dust deposition flux over the Amazon? This can be addressed by a trajectory study as well and would be a worthy contribution to the scientific discussion on the fertilization impact of northern African dust sources.

(3) Was the dust mass per source region calculated for all identified dust storms originating from the corresponding location or for a selected number only? This refers to line 18-20, page 7. Please clarify. Same for the numbers provided in the results section: Are these based on all identified dust storms? Which quantities are put in relation?

(4) Can you please explain in more detail how the dust mass fractions were calculated (refers to line 9-12, page 9)? Providing the fraction implies you know the annual total - or at least did some assumption. Which mass fractions are related? Please clarify.

Minor comments:

p 5, l 15-16 It is absolutely reasonable to limit the calculation to the ten strongest dust storms. Can you add a brief statement on the representativeness of the results despite this selection is made?

p 7, l 9-14 Please consider referring to a map illustrating the location of the listed source

regions.

p 13, l 6 There are studies highlighting the Sudan as active dust source. E.g. Schepanski et al., (Geophy. Res. Lett. 2007, J. Geophys. Res. 2009), and Formenti et al., (Atmos. Chem. Phys., 2011).

---

## Referee Comment (RC2) · Ian Ashpole (Referee) · 10 May 2019

General comments

This is an enjoyable and insightful paper that combines multiple different satellite datasets to enhance our knowledge of the geomorphology of North African dust sources and estimate how much dust is emitted from different regions. As well as being of scientific value in themselves, the results are of significance for our understanding of the fertilization link between Saharan dust and the Amazon and this case is strongly made in the paper introduction, discussion and conclusion. However, I do feel that the methods need greater clarification/explanation, and that in some places a

mention of the methodological/data limitations is needed. In my opinion, this paper will be well worthy of publication once a few issues are addressed.

My review is structured as follows: I first explain my major comment about the paper. I then outline a series of specific comments, section-by-section, which are less minor (some of these will likely be addressed in the process of addressing my major comment), before suggesting a few small technical corrections at the end.

————

Major Comment

My only major question/comment refers to your methods and is more a question of greater explanation/clarification needed, as opposed to me questioning the validity of what you have done (the method seems great to me, if I understand it correctly).

Basically, I'm not entirely clear how you get to the final results that are presented in Tables 1 and 2. From several re-reads of the methods, it is my understanding that you only classify the source geomorphology (Section 2.3) of the 20 dust events that you study in detail in Section 2.2 (using VIIRS)? Yet, in Table 1, you give the classification of 3512 sources. Does this mean that the 20 events studied originated from 3512 specific point sources in total; or does this number come from all of the events identified in the SEVIRI data (Section 2.1)? If the latter, how do you know the geomorphology of the point sources that contribute to the events that were *not* studied using the VIIRS data in Section 2.2 and Sentinel data in Section 2.3? (A related point: you don't state anywhere in Section 2.1 that you identify the actual sources of events identified using SEVIRI data, just that the events were "identified". If you did backtrack these events to sources, you should say so.)

I *think* the answer to the above is in the first couple of sentences of Section 2.6, but I am not 100% sure: "Analysis of the dust source locations from the 10 largest dust storms in each dust season showed that a single geomorphological source dominated

in most areas. On the basis of this observation, 16 individual dust source regions were identified and mapped." Does this mean, that all (or most of) the events identified with SEVIRI (Section 2.1) originated from within the 16 broader regions that are outlined in Figure 3? And because you had studied in detail events that originated from these sources in the course of Sections 2.2 and 2.3, you were able to deduce the geomorphology of the whole broader area? So, for example, if an event detected in SEVIRI that was not one of the largest 20 originated from within the boundaries of the Bodélé, it was assumed to have a source geomorphology of paleolake? If I *am* correct about this, then I am confused as to how you get stats in Table 1 for geomorphology types of "Sand deposits" through to "Anthropogenic", since all of the 16 broad regions are classified as "Alluvial deposits" and/or "Paleolakes"*. But then, some of the pins in Figure 3 (which are the sources of the 20 largest events) are colour coded to represent "Sand deposits" etc. So, do the figures in Table 1 actually only come from these top 20 events? Hopefully you can see why I am confused!

*Relatedly: 4 of these broad regions weren't analysed as part of Sections 2.2 and 2.3 as none of the 20 biggest events originated from them: At what stage, then, did you classify their geomorphology? I think this is implied in "mapping" stage of the quoted text from the start of Section 2.6, but how you did this mapping isn't clear.

————

Specific comments

(*Suggested text modifications for each section are in italics*)

General: It would be helpful to clarify/be consistent with your use of the term "source" and variants ("point source", "source region", etc). In some cases you use "source" to refer to the specific point sources and features identified in Sections 2.2 and 2.3, and in other cases you use it to refer to the 16 broader source regions from Figure 3. Sometimes you use "point source" and "source regions" to refer to these, but not always.

Abstract

Arguably, the first 3 sentences set this up to be a paper about fertilization, but this isn't really its focus – Amazon fertilization can only really be commented on from your (albeit quite significant) results. Can I suggest shifting these sentences to the end of the abstract/de-emphasizing the fertilization angle?

1 Introduction

P1, L20: Mineral dust is likely important to more parts of the earth system than just climate (i.e. biogeochemical cycles, as you emphasise this paper). You could broaden this opening statement a little to reflect this.

P2, L14: "While the major dust source areas within the Sahara and the Sahel have been identified. . .less is known about the geomorphology of these sources". True, but some studies (e.g. Ashpole and Washington 2013, Crouvi et al 2012) have considered this (AW13 identifies several alluvial features as dust sources in the central and western Sahara) and could be mentioned here?

P2, L28: ". . .as well as the total dust mass emitted per class and dust source region are calculated". Should "calculated" be changed to "estimated", since the "calculations" contain a terms with a lot of potential error/unknowns?

2.1. Detecting dust storms

Did you detect the sources of all dust events that were identified? You don't explicitly say this, but it seems that you must have done for the later methods to make sense?

P5, L6-7: "By visually interpreting images, all dust events of the 2015-2017 boreal winter dust seasons were identified". Did you have some subjective threshold of what was and wasn't a dust event? From experience of working with these data I know that there are sometimes minor dust plumes that are hard to reliably trace to a specific source area. Similarly, how about when detection/interpretation/tracking was complicated by the presence of clouds? I appreciate that you do the best job you can with the data at

hand and that no analysis is perfect, but I think it is important to note somewhere if you actually just identified, for example, all dust events that could be tracked to a source with confidence, since this will introduce an element of uncertainty (albeit probably small) into your results.

2.2 Identifying dust sources

P5, L14-15 (and also Figure 2): "Dust sources were identified manually using visual interpretation, by tracking the individual plumes back to their first point of occurrence (illustrated in Figure 2)". I think this needs a little more explanation/clarification: The VIIRS data just show you a dust plume at a single point in time. How can you be *sure* what the first point of occurrence is? Is it taken as the most upwind point of the plume (with the upwind direction deduced from watching plume evolution in the SEVIRI data)? Additionally, in the example of Figure 2, you seem to have attributed the plume to 5 specific source locations, but what about the spaces in between these points, which also seem to be a shade of pink? Or is the whole upwind edge actually taken as the source location and the black points are just illustrative? OR, is this stage just narrowing down the suspected first points of occurrence so that you know what area to study for likely emissive features in a later step? Further explanation/clarification would be helpful here!

P5, L16: It would be helpful to clarify whether the selection of the 10 largest dust storms per season was qualitative, or whether some other quantitative means was used to pick them (such as area covered, based on the step outlined in Section 2.4?).

2.3 Classifying geomorphology of dust sources

General: Is it possible to include example images (or a case study) of sources meeting each classification as an additional figure (or supporting material)? I think this would help non-expert readers to understand (and visualise) the different source types.

P6, L11-13: "To account for the spatial resolution differences between VIIRS and

Sentinel-2, point sources were classified as the most prominent feature and/or most common geomorphic class within 750 m of the point source." I find the wording a little confusing here. Does the second "point source" refer to the original point identified in the VIIRS data?

2.4 Determining dust plume area

General:

1) What happens when dust plumes detected & measured in VIIRS are the result of several smaller plumes from different sources merging? How do you apportion the plume between the different sources? Or is this never an issue?

2) A problem with using VIIRS to determine dust plume area as opposed to SEVIRI is that it only gives one snapshot per day of dust plumes. Given that the size of plumes will change due to transport and advection, this must introduce a bias in your results, since if VIIRS observed the plumes at a different time of day they would be smaller or larger? This has knock-on effects for average plume AOD calculations and dust mass calculations. You are limited in what you can do by the data at hand, but this seems important to acknowledge. Relatedly, what time of day is the VIIRS observation actually made, and how does this relate to the modal emission time for the region, which I think is during the morning in this season (based on the work of e.g. Schepanski et al. 2009 & others)?

P6, L23-24: "dust plumes could be outlined and their size calculated". For clarification, this enables you to treat each individual dust plume separately and assign that dust plume area to certain source?

2.5 Measuring average aerosol optical depth per plume

General: You may have clarified this in addressing my major comment, but do you get the average AOD of every dust plume identified in Section 2.1, or just the subset studied in Sections 2.2 – 2.3?

General: How do you actually go about calculating the average AOD per plume? Is it a case of overlapping the VIIRS and MODIS data, getting all AOD values that overlap with flagged dust plumes from Section 2.4, and calculating the arithmetic mean (or median) for each individual dust plume? How do you deal with the resolution mismatch between the MODIS data (10 km) and VIIRS data (0.75 km)?

P6, L31: "MODIS monitors AOD on a daily basis". Is there any temporal offset between the MODIS and VIIRS data, or are the sensors carried on the same satellite? This seems important to me because if the plume area/mean AOD are obtained at different points in the plume lifetime, its location may have changed, which would be problematic if using VIIRS pixels to get AOD values from MODIS data. . .

2.6 Calculating dust mass per dust source

Suggested modification for section title: "Estimating emitted dust mass per dust source region"? Terms in your calculation contain a lot of potential error/uncertainty (plume size, mean AOD, extinction coefficients. . .), so this is really more of a rough estimate. Also, you seem to consider dust mass for broader source regions as opposed to individual point sources, which is what the term "source" also occasionally refers to in this paper.

P7, L9-17: This paragraph seems like it could be better placed in its own sub-section, explaining how you pull the methods of Section 2.1 – 2.5 together to create a set of dust plumes per broad source region, as that step (if I understand your methods correctly) seems like it needs greater emphasis. However, this may not be the case depending on how you address my major comment.

P7, L9-10: "Analysis. . .showed that a single geomorphological source dominated in most areas". Do you mean single geomorphological source class?

P7, L10: "On the basis of this observation, 16 individual dust source regions were identified and mapped". What does "and mapped" mean? Does this refer to digital

boundary creation or some such to help with categorising the source area of all of the individual plumes detected in Section 2.1? I think clarification/clearer explanation is needed in this section.

P7, L14-15: "Each dust plume was then assigned a geomorphic class based on the region it came from". I'm confused. By "each dust plume", are you referring to all the plumes detected in Section 2.1? If so, you didn't state in that section that you identified sources. I am guessing this must be the case because (I think?) that you also have the size and mean AOD for all these plumes. . .

P7, L15-17: "The accuracy of this methodology was evaluated by randomly selecting three additional days, analysing the geomorphic class of each source, and determining what percentage of these dust sources coincided with that defined by the map of predominant dust geomorphologies." Suggested addition for clarity: ". . .analysing the geomorphic class of each detected dust source. . ." Additionally: does it make sense to move your statements about the accuracy assessment results from the end of Section 3 (P9, L9-11) to here? They don't add anything to the overall paper results, and the current wording here leaves me asking the question "well. . .how accurate did your methodology turn out to be?!"

P7, L19: ". . .whereby dust mass is calculated. . ." Again, can I suggest calculated be substituted for estimated? Can I also suggest a statement be made about the reliability of these results, given all of the (presumably quite uncertain and highly variable) parameters in Kaufman's equations, as well as the fact that you are using a plume average AOD and plume size calculation which are taken from one snapshot through the dust event's lifetime (and these values could therefore be quite different if calculated using observations at a different time of day).

P8, L3: "although it should be noted that the equation has an uncertainty of $\pm 30\%$ for AOD values between 0.2 and 0.4." Ben-Ami et al. also note that "in cases when total AOD > 0.4, the error may be larger" – this should be added.

3 Results

General: In Table 2, you only seem to attribute dust mass to events originating from within the 16 broad source regions outlined in Figure 3. This figure shows that a small number of the point sources identified during the 20 largest dust storms that you analysed in Section 2.2 fell outside of these regions (e.g. in Egypt) and, presumably, at least a small portion of the events identified in SEVIRI data (Section 2.1) did too. If I'm right about your methodology and you calculated dust mass for all plumes, this means your "Total" mass stats (and statements about this) are missing emission from these events? Even if this is a minor proportion of the total mass emitted from the 16 main source regions, it still seems worthy of mention.

P8, L19: Can you give mean AOD for the other source regions, in addition to the Bodélé Depression? It would be interesting to see whether there are significant differences between sources. For example, some source areas (such as Sudan and Ennedi, both alluvial) seem to be frequently active, but emit a comparatively small amount of dust (and the converse is true for, e.g. Algeria – paleolake). It would be interesting to know if this is a function of differences in mean AOD or plume size. Given your comments about differences in the biogeochemical make up of dust from different source geomorphologies, it seems plausible that this might in some way affect the observed AOD values from specific source types too. Alternatively, it could say something about differences in erosivity/erodibility in these regions. Either way, worthy of comment as this is useful knowledge for remote sensing/dust modelling communities (as well as others!)

P9, L4-5 & Table 2: A reminder (possibly just in Table 2 caption) that South Air and Erg Chech dust mass was apportioned 50/50 to the paleolake and alluvial deposit categories would be useful, as it took me a while to work out how exactly you got to the 64% & 36% numbers.

P9, L11-12: "The ten largest dust storms per season emitted 37.6% of the total dust

mass." What proportion of the total detected dust storms did the ten largest dust storms per season account for? This would be useful to know to contextualise the 37.6% figure.

————

Technical corrections (typos, citations, etc)

P5, L4: add a reference to Figure 2 (as this visualises pink dust)?

P5, L5: "…and blue the 10.8 $\mu$m channel". I suggest the following addition for clarity (since this is not a BTD): "…and blue the brightness temperature (BT) of the 10.8 $\mu$m channel".

P5, L8: "[the high temporal resolution of SEVIRI] makes it suitable for observing dust storms". I suggest the following addition: "[the high temporal resolution of SEVIRI] makes it suitable for observing the evolution of dust storms". This is the strength of the higher temporal resolution data; you can still "observe" dust storms (or dust plumes, at least) with lower temporal resolution data.

P5, L8-9: "…its low spatial resolution means it is less suited to determining the location and nature of the sources". I suggest the following addition: "…its low spatial resolution means it is less suited to determining the precise location and nature of the sources". The strength of the higher spatial resolution is the precision it affords, since SEVIRI can give the correct location to a scale of 3 km, no?

P8, L13: "During the 2015-2017 winter dust season between 82.3 - 127.0 Tg of dust was emitted". You should include a reference to Table 2 here.

P12, L15: "N and P macronutrients". Expand abbreviations/chemical symbols?

---

## Author Comment (AC1) · 21 Jun 2019

**Responses to reviewer #1:**

Please find our responses to the review below. Original review is displayed in black, our responses in blue.

The study is motivated by the question, how much phosphorus is supplied to the Amazon by North African dust sources. The authors examine the source regions for winter time dust storms (as this is the season when dust is transported towards the Amazon) for three consecutive years (2015-2017). Geomorphological characteristics of the source region were determined for the ten strongest dust storms per winter season. The manuscript is well written and I enjoyed reading it. Nevertheless, I have a few questions I would like the authors to address before publication

Response: We would like to thank the reviewer for the assessment of the paper and for the constructive feedback and helpful questions.

General comments: (1) The question I am most curious to know the answer remains somewhat unanswered: Can you estimate how much phosphorus is supplied to the Amazon by the individual dust source regions? E.g., Schepanski et al., Atmos. Chem. Phys. (2009) examined by means of a model simulation the contribution of dust emitted from the Bodélé Depression to the atmospheric dust burden over the Gulf of Guinea and tropical Atlantic showing a contribution of up to 60% over the Equator region. But it remains unclear, how much dust passes through the tropical rain belt. I think it may be worth considering adding a trajectory analysis here showing how many dusty trajectories originating from an emitting dust source actually reach the Amazon without being affected by rainfall or clouds.

Response: We fully agree with the reviewer that this question of 'how much phosphorus is supplied to the Amazon by the individual dust source regions?' is an important one which needs answering. In order to do so, however, two extra key pieces of information are needed. The first is how much phosphorus is in the dust derived from the different types of sources that we identify, the second is how much of this dust reaches the Amazon (i.e. trajectory analysis, as mentioned).

All studies on the (fertilisation) effect of dust transport to the Amazon employ the phosphorus concentration of dust from the Bodélé Depression paleolake sediments; we do not have any information on how much phosphorus is in dust from other sources at present (e.g. other paleolakes or the paleorivers we have identified). The results from this paper show that the Bodélé is not representative of the diversity of surfaces producing dust, and so more data is needed on the phosphorus concentration in the different dust sources before this question can be answered accurately. We have sampled dust from various paleoriver and paleolake systems throughout much of Northern Africa, which we are currently analysing to get a better understanding of the phosphorus concentration in Northern African dust from the individual sources in different regions. This research will be published in a follow up paper that builds on the research presented in this paper.

With regards to the trajectory analysis we again agree with the reviewer – this is a good idea. To achieve this, we are currently developing a new remote sensing trajectory analysis methodology using MODIS AOD to track dust from source to sink. This will form the basis of another follow up paper.

While we are striving to answer the questions raised by the reviewer in due course, by carrying out extensive fieldwork, lab work, and developing and testing this new remote sensing methodology for dust tracking, this involves a considerable amount of research which will be published in papers that follow on from this one. If all this research were to be combined it would form a huge paper, with multiple methods and conclusions that would bury some of the important findings.

(2) Related to question (1), how many of the identified dust sources actually contribute to the dust deposition flux over the Amazon? This can be addressed by a trajectory study as well and would be a worthy contribution to the scientific discussion on the fertilization impact of northern African dust sources

Response: This follows on from the previous question. We agree this would be a very worthy contribution to the scientific discussion, however as we would like to tackle this issue using a new remote sensing methodology in order to answer the question most accurately, we believe this would be better suited for a follow-up manuscript, to eventually be combined with our geochemical analysis of dust samples that we have collected from paleorivers and paleolakes throughout northern Africa.

(3) Was the dust mass per source region calculated for all identified dust storms originating from the corresponding location or for a selected number only? This refers to line 18-20, page 7. Please clarify. Same for the numbers provided in the results section: Are these based on all identified dust storms? Which quantities are put in relation?

Response: The dust mass per source region was calculated for all identified storms. Upon reflection this was not clearly explained (it was also mentioned by reviewer #2), and therefore we have revised the methodology section. It is now clearly stated that the selected number of dust storms (20 largest) only pertains to identifying the location of dust point sources and the classification of their geomorphology (sections 2.2 and 2.3). This has also been better clarified in the results/figures/tables.

(4) Can you please explain in more detail how the dust mass fractions were calculated (refers to line 9-12, page 9)? Providing the fraction implies you know the annual total - or at least did some assumption. Which mass fractions are related? Please clarify.

Response: The dust mass fraction is calculated as the ratio between the dust mass emitted during the 10 largest dust storms per season and the total dust mass emitted during the wintertime seasons. The total Tg emitted during the 2015/2016 and 2016/2017 winter dust seasons (Dec/Jan/Feb) was 82.3 - 127.0, which can be found in Table 2. The ten largest dust storms of both seasons emitted 30.95-47.75 Tg. This is how the 37.6% on line 12, page 9 (original manuscript) was calculated. As it was not clear that the values in Table 2 corresponded to all dust storms in the season, this has been clarified in both methodology and in the Table title. As per suggestion of reviewer #2, we have now split up this figure per season.

p 5, l 15-16 It is absolutely reasonable to limit the calculation to the ten strongest dust storms. Can you add a brief statement on the representativeness of the results despite this selection is made?

Response: Only with regards to the exact location of individual dust point sources and their geomorphology (methodology section 2.2 and 2.3) did we limit our methodology to the ten strongest dust storms. All other analyses were carried out for all dust storms in the seasons. Due to time constraints and labour-intensive methodology of identifying point sources and classifying

geomorphology per point source we decided to limit to the ten largest dust storms per season. The representativeness of this is shown in the accuracy assessment, and the results of this can be found at lines 9-12, page 9 of the original manuscript. Nevertheless, it seems this could do with better explanation; as such we have combined these sections in a new paragraph to make it clearer (last paragraph in section 2.3 of revised manuscript).

p 7, l 9-14 Please consider referring to a map illustrating the location of the listed source regions.

Response: The map presented (in the results section of the original manuscript) illustrates the location of the listed source regions; on reflection this map was indeed better placed in the methodology section. We have moved the map and added a sentence referring to the map.

p 13, l 6 There are studies highlighting the Sudan as active dust source. E.g. Schepanski et a, (Geophy. Res. Lett. 2007, J. Geophys. Res. 2009), and Formenti et al., (Atmos. Chem. Phys., 2011).

Response: We thank the reviewer for bringing this to our attention - this was indeed an error. There are a couple of studies referring to Sudan, however these are nearly all pertaining to northern Sudan/Nubian desert. As you mention, Schepanski (2009) is the exception in naming central Sudan. We have updated this sentence to reflect we are referring to central/southern Sudan and have added Schepanski (2009).

---

## Author Comment (AC2) · 21 Jun 2019

**Responses to reviewer #2 (Ian Ashpole):**

Please find our responses to the review by Dr. Ian Ashpole below. Original review is displayed in black, our responses in blue.

General comments: This is an enjoyable and insightful paper that combines multiple different satellite datasets to enhance our knowledge of the geomorphology of North African dust sources and estimate how much dust is emitted from different regions. As well as being of scientific value in themselves, the results are of significance for our understanding of the fertilization link between Saharan dust and the Amazon and this case is strongly made in the paper introduction, discussion and conclusion. However, I do feel that the methods need greater clarification/explanation, and that in some places a mention of the methodological/data limitations is needed. In my opinion, this paper will be well worthy of publication once a few issues are addressed.

Response: We would like to sincerely thank Ian Ashpole for the thorough review and we greatly appreciate the valuable questions and suggestions.

My review is structured as follows: I first explain my major comment about the paper. I then outline a series of specific comments, section-by-section, which are less minor (some of these will likely be addressed in the process of addressing my major comment), before suggesting a few small technical corrections at the end.

My only major question/comment refers to your methods and is more a question of greater explanation/clarification needed, as opposed to me questioning the validity of what you have done (the method seems great to me, if I understand it correctly).

Basically, I'm not entirely clear how you get to the final results that are presented in Tables 1 and 2. From several re-reads of the methods, it is my understanding that you only classify the source geomorphology (Section 2.3) of the 20 dust events that you study in detail in Section 2.2 (using VIIRS)? Yet, in Table 1, you give the classification of 3512 sources. Does this mean that the 20 events studied originated from 3512 specific point sources in total; or does this number come from all of the events identified in the SEVIRI data (Section 2.1)? If the latter, how do you know the geomorphology of the point sources that contribute to the events that were *not* studied using the VIIRS data in Section 2.2 and Sentinel data in Section 2.3? (A related point: you don't state anywhere in Section 2.1 that you identify the actual sources of events identified using SEVIRI data, just that the events were "identified". If you did backtrack these events to sources, you should say so.)

Response: This is a very valid point. It seems the methodology section could do with greater explanation/clearer wording. We used the 20 largest dust events to determine dust point sources and the geomorphology of these dust point sources (sections 2.2 & 2.3), all other analyses are done on ALL dust events in the wintertime seasons. In summary we use the dust point sources from the 20 largest dust storms and their geomorphology to derive the major dust sources regions. Then we assign each dust source region a predominant geomorphology. Dust plumes (all in the season) are allocated to a dust source region and assigned the predominant geomorphic class of that region. We have now added sentences in all methodology subsections to clarify this. The results section has also been updated to clarify this. It is correct that the 20 largest dust events had 3512 specific point sources in total. We did not identify the point sources of all the events identified using SEVIRI, this was done only on the 20 largest dust events.

I *think* the answer to the above is in the first couple of sentences of Section 2.6, but I am not 100% sure: "Analysis of the dust source locations from the 10 largest dust storms in each dust season showed that a single geomorphological source dominated in most areas. On the basis of this observation, 16 individual dust source regions were identified and mapped." Does this mean, that all (or most of) the events identified with SEVIRI (Section 2.1) originated from within the 16 broader regions that are outlined in Figure 3? And because you had studied in detail events that originated from these sources in the course of Sections 2.2 and 2.3, you were able to deduce the geomorphology of the whole broader area?

Response: Yes, this is correct. The dust source regions and their predominant geomorphology were determined from the location and geomorphological class of the individual dust point sources. All dust plumes identified with SEVIRI were assigned to one of the 16 broader regions based on the location of the upwind edge of the plume, and then assigned the predominant geomorphology of that region. We have revised our methodology section to make this clearer.

So, for example, if an event detected in SEVIRI that was not one of the largest 20 originated from within the boundaries of the Bodélé, it was assumed to have a source geomorphology of paleolake? If I *am* correct about this, then I am confused as to how you get stats in Table 1 for geomorphology types of "Sand deposits" through to "Anthropogenic", since all of the 16 broad regions are classified as "Alluvial deposits" and/or "Paleolakes"*. But then, some of the pins in Figure 3 (which are the sources of the 20 largest events) are colour coded to represent "Sand deposits" etc. So, do the figures in Table 1 actually only come from these top 20 events? Hopefully you can see why I am confused!

Response: The figures in Table 1 refer to the point sources of the 20 largest dust events. From the 20 largest dust events a total of 3512 individual dust point sources were identified. We have added this information in the first sentence of the results section. All these point sources were classified using the geomorphic classification scheme (e.g. Paleolake, Sand Deposit, Anthropogenic etc.). As we found that dust sources in similar regions were largely derived from the same dust source, the dust plumes which were not part of the 20 largest dust events were allocated the predominant geomorphology of the region it came from. So yes, an event detected in SEVIRI that was not one of the largest 20, which originated from within the boundaries of the Bodélé, was assumed to have a source geomorphology of paleolake. We have re-written our explanation of this aspect of the methodology to make this clearer.

*Relatedly: 4 of these broad regions weren't analysed as part of Sections 2.2 and 2.3 as none of the 20 biggest events originated from them: At what stage, then, did you classify their geomorphology? I think this is implied in "mapping" stage of the quoted text from the start of Section 2.6, but how you did this mapping isn't clear.

Response: When we assigned dust plumes to a region we determined 4 additional broad regions, which were not part of the 20 biggest events. At this stage we classified the geomorphology of these 4 regions using the same methods as the 20 largest dust storms. This was part of section 2.3. We have revised our methodology to include this information and clarify the process.

General: It would be helpful to clarify/be consistent with your use of the term "source" and variants ("point source", "source region", etc). In some cases you use "source" to refer to the specific point

sources and features identified in Sections 2.2 and 2.3, and in other cases you use it to refer to the 16 broader source regions from Figure 3. Sometimes you use "point source" and "source regions" to refer to these, but not always.

Response: Thanks for this excellent observation, this is indeed confusing, we have corrected this such that we refer to either 'point source' or 'source region' in all cases.

Abstract Arguably, the first 3 sentences set this up to be a paper about fertilization, but this isn't really its focus – Amazon fertilization can only really be commented on from your (albeit quite significant) results. Can I suggest shifting these sentences to the end of the abstract/de-emphasizing the fertilization angle?

Response: The problem with this suggestion is that we only look at winter dust plumes because we are interested in dust that goes to the Amazon. If we do this, the reader will wonder why we are only looking at winter plumes, and not a full year. Given this we feel we need to explain the Amazon fertilisation angle up front. As such we have decided to not change the abstract as requested. We have, however, made it clearer that the reason why we only look at winter plumes is because we are ultimately interested in Amazon dust fertilisation.

1 Introduction P1, L20: Mineral dust is likely important to more parts of the earth system than just climate (i.e. biogeochemical cycles, as you emphasise this paper). You could broaden this opening statement a little to reflect this.

Response: The sentence has been replaced by: 'Mineral dust is an important component of the Earth system, affecting radiative forcing, cloud properties, and playing a key role in terrestrial, oceanic, atmospheric, and biogeochemical exchanges (Harrison et al., 2001; Jickells et al., 2005; Mahowald et al., 2010).'

P2, L14: "While the major dust source areas within the Sahara and the Sahel have been identified. . .less is known about the geomorphology of these sources". True, but some studies (e.g. Ashpole and Washington 2013, Crouvi et al 2012) have considered this (AW13 identifies several alluvial features as dust sources in the central and western Sahara) and could be mentioned here?

Response: We have altered the sentence to: 'While the major dust source areas within the Sahara and the Sahel have been identified …, only few studies classify the geomorphology of these sources (Crouvi et al., 2012; Ashpole & Washington, 2013), and knowledge of the relative importance of the major dust regions and is lacking.'

P2, L28: ". . .as well as the total dust mass emitted per class and dust source region are calculated". Should "calculated" be changed to "estimated", since the "calculations" contain a terms with a lot of potential error/unknowns?

Response: Agreed estimated is the better term, 'calculations' has been replaced by 'estimated'.

2.1. Detecting dust storms
Did you detect the sources of all dust events that were identified? You don't explicitly say this, but it seems that you must have done for the later methods to make sense?

Response: We did not detect point sources for all dust events. We identified the region it originated from based on the upwind edge of the plume. This information has been added.

P5, L6-7: "By visually interpreting images, all dust events of the 2015-2017 boreal winter dust seasons were identified". Did you have some subjective threshold of what was and wasn't a dust event? From experience of working with these data I know that there are sometimes minor dust plumes that are hard to reliably trace to a specific source area. Similarly, how about when detection/interpretation/tracking was complicated by the presence of clouds? I appreciate that you do the best job you can with the data at hand and that no analysis is perfect, but I think it is important to note somewhere if you actually just identified, for example, all dust events that could be tracked to a source with confidence, since this will introduce an element of uncertainty (albeit probably small) into your results.

Response: SEVIRI was used as quick initial visual tool to observe dust storms, the actual analysis of dust mass was done using VIIRS for which the thresholds of your (Ashpole & Washington, 2012) automated dust detection technique are used (Eq. 1). Detecting dust sources under clouds is impossible with the Dust RGB technique, thus these were not included in the study. We have added this information: 'The individual dust plumes were tracked upwind back to their first point of occurrence (see Figure 2), except for those forming under clouds (dust storms with sources under clouds were visible on approximately 2 days per month and this was thus not a significant problem).'

2.2 Identifying dust sources P5, L14-15 (and also Figure 2): "Dust sources were identified manually using visual interpretation, by tracking the individual plumes back to their first point of occurrence (illustrated in Figure 2)". I think this needs a little more explanation/clarification: The VIIRS data just show you a dust plume at a single point in time. How can you be *sure* what the first point of occurrence is? Is it taken as the most upwind point of the plume (with the upwind direction deduced from watching plume evolution in the SEVIRI data)? Additionally, in the example of Figure 2, you seem to have attributed the plume to 5 specific source locations, but what about the spaces in between these points, which also seem to be a shade of pink? Or is the whole upwind edge actually taken as the source location and the black points are just illustrative? OR, is this stage just narrowing down the suspected first points of occurrence so that you know what area to study for likely emissive features in a later step? Further explanation/clarification would be helpful here!

Response: The first point of occurrence is indeed the most upwind point of the plume. We have added this information in the manuscript for clarification (in section 2.2). The spaces in between the points are a lighter shade of pink showing dispersed dust.

P5, L16: It would be helpful to clarify whether the selection of the 10 largest dust storms per season was qualitative, or whether some other quantitative means was used to pick them (such as area covered, based on the step outlined in Section 2.4?).

Response: It is based on areal coverage. We have replaced 'spatial extent' for 'areal coverage'.

2.3 Classifying geomorphology of dust sources General: Is it possible to include example images (or a case study) of sources meeting each classification as an additional figure (or supporting material)? I think this would help non-expert readers to understand (and visualise) the different source types.

Response: This has been added as an Appendix and a reference to the appendix has been included.

P6, L11-13: "To account for the spatial resolution differences between VIIRS and Sentinel-2, point sources were classified as the most prominent feature and/or most common geomorphic class within 750 m of the point source." I find the wording a little confusing here. Does the second "point source" refer to the original point identified in the VIIRS data?

Response: Yes. This has been changed into 'point sources were classified as the most prominent feature and/or most common geomorphic class within 750 m of the point location determined with VIIRS'

2.4 Determining dust plume area

General: 1) What happens when dust plumes detected & measured in VIIRS are the result of several smaller plumes from different sources merging? How do you apportion the plume between the different sources? Or is this never an issue?

Response: Most often individual point sources merge into one dust plume (as can be seen in Figure 2, 5 point sources – 1 plume). We do not apportion the dust mass to a single dust point source, only to dust plumes. The dust plumes could be attributed to one source region with a known source geomorphology.

2) A problem with using VIIRS to determine dust plume area as opposed to SEVIRI is that it only gives one snapshot per day of dust plumes. Given that the size of plumes will change due to transport and advection, this must introduce a bias in your results, since if VIIRS observed the plumes at a different time of day they would be smaller or larger? This has knock-on effects for average plume AOD calculations and dust mass calculations. You are limited in what you can do by the data at hand, but this seems important to acknowledge. Relatedly, what time of day is the VIIRS observation actually made, and how does this relate to the modal emission time for the region, which I think is during the morning in this season (based on the work of e.g. Schepanski et al. 2009 & others)?

Response: VIIRS mean overpass time is 1.30pm local time, which coincides with MODIS Aqua data. The difference of the actual observation time between VIIRS and MODIS was <20 minutes. We have added the overpass times and the difference in the manuscript. Most dust storms start (first observation) between 06.00-09.00 and 09.00-12.00, with a few sources starting emission between 12.00-15.00 (Schepanski et al., 2009). However, we noticed that most sources emit for several hours and thus even if they initiate early in the morning they will still be detected by VIIRS. We found that on only 1-2 days per season a dust storm initiated after 13.30 and were hence not included in the analysis (which we also added to the manuscript). Our other AOD option would have been to use MODIS Terra, at a mean local overtime pass of 10.30AM. The downsides of Terra compared to Aqua would be that we would have missed dust storms starting after 10.30AM, we would have underestimated dust storms which were still emitting between 10.30-13.30 (this was especially common for larger dust storms), and we would have to use SEVIRI data with lower spatial resolution as the overpass time would not coincide with VIIRS. As such, we decided that the combination of VIIRS and MODIS Aqua was better suited.

P6, L23-24: "dust plumes could be outlined and their size calculated". For clarification, this enables you to treat each individual dust plume separately and assign that dust plume area to certain source?

Response: Yes. This has been clarified (section 2.3).

2.5 Measuring average aerosol optical depth per plume General: You may have clarified this in addressing my major comment, but do you get the average AOD of every dust plume identified in Section 2.1, or just the subset studied in Sections 2.2 – 2.3?

Response: Has been addressed in response to major comment, but indeed, we get average AOD for all dust plumes identified in section 2.1.

General: How do you actually go about calculating the average AOD per plume? Is it a case of overlapping the VIIRS and MODIS data, getting all AOD values that overlap with flagged dust plumes from Section 2.4, and calculating the arithmetic mean (or median) for each individual dust plume? How do you deal with the resolution mismatch between the MODIS data (10 km) and VIIRS data (0.75 km)?

Response: We determine a plume boundary outline from VIIRS, this is overlapped on the MODIS data, and the arithmetic mean is calculated using statistics analysis in ENVI based on the overlapped region. The plume boundary is a georeferenced vector layer, such that resolution mismatch does not pose a problem.

P6, L31: "MODIS monitors AOD on a daily basis". Is there any temporal offset between the MODIS and VIIRS data, or are the sensors carried on the same satellite? This seems important to me because if the plume area/mean AOD are obtained at different points in the plume lifetime, its location may have changed, which would be problematic if using VIIRS pixels to get AOD values from MODIS data. . .

Response: As mentioned before, VIIRS and MODIS Aqua have the same mean overpass time. This information has been added.

2.6 Calculating dust mass per dust source

Suggested modification for section title: "Estimating emitted dust mass per dust source region"? Terms in your calculation contain a lot of potential error/uncertainty (plume size, mean AOD, extinction coefficients. . .), so this is really more of a rough estimate. Also, you seem to consider dust mass for broader source regions as opposed to individual point sources, which is what the term "source" also occasionally refers to in this paper.

Response: We agree with the suggestion, this has been modified.

P7, L9-17: This paragraph seems like it could be better placed in its own sub-section, explaining how you pull the methods of Section 2.1 – 2.5 together to create a set of dust plumes per broad source region, as that step (if I understand your methods correctly) seems like it needs greater emphasis. However, this may not be the case depending on how you address my major comment.

P7, L9-10: "Analysis. . .showed that a single geomorphological source dominated in most areas". Do you mean single geomorphological source class?

Response: Yes, this has been corrected.

P7, L10: "On the basis of this observation, 16 individual dust source regions were identified and mapped". What does "and mapped" mean? Does this refer to digital boundary creation or some such to help with categorising the source area of all of the individual plumes detected in Section 2.1? I think clarification/clearer explanation is needed in this section.

Response: 'and mapped' refers to 'outlined on a map'. As mentioned before, this part has been re-written and moved to section 2.3.

P7, L14-15: "Each dust plume was then assigned a geomorphic class based on the region it came from". I'm confused. By "each dust plume", are you referring to all the plumes detected in Section 2.1? If so, you didn't state in that section that you identified sources. I am guessing this must be the case because (I think?) that you also have the size and mean AOD for all these plumes. . .

Response: Yes, 'each dust plume' refers to all plumes. As per the major question/comment this has been clarified.

P7, L15-17: "The accuracy of this methodology was evaluated by randomly selecting three additional days, analysing the geomorphic class of each source, and determining what percentage of these dust sources coincided with that defined by the map of predominant dust geomorphologies." Suggested addition for clarity: ". . .analysing the geomorphic class of each detected dust source. . ." Additionally: does it make sense to move your statements about the accuracy assessment results from the end of Section 3 (P9, L9-11) to here? They don't add anything to the overall paper results, and the current wording here leaves me asking the question "well. . .how accurate did your methodology turn out to be?!"

Response: Yes, that indeed makes more sense. We have moved it.

P7, L19: ". . .whereby dust mass is calculated. . ." Again, can I suggest calculated be substituted for estimated? Can I also suggest a statement be made about the reliability of these results, given all of the (presumably quite uncertain and highly variable) parameters in Kaufman's equations, as well as the fact that you are using a plume average AOD and plume size calculation which are taken from one snapshot through the dust event's lifetime (and these values could therefore be quite different if calculated using observations at a different time of day).

Response: Calculated has been substituted for estimated. We have added that the daily temporal resolution of MODIS/VIIRS is a limitation to the study in section 2.5 and put greater emphasis on the uncertainty of the Kaufman equation in section 2.6.

P8, L3: "although it should be noted that the equation has an uncertainty of ±30% for AOD values between 0.2 and 0.4." Ben-Ami et al. also note that "in cases when total AOD > 0.4, the error may be larger" – this should be added.

Response: This has been added.

3 Results

General: In Table 2, you only seem to attribute dust mass to events originating from within the 16 broad source regions outlined in Figure 3. This figure shows that a small number of the point sources

identified during the 20 largest dust storms that you analysed in Section 2.2 fell outside of these regions (e.g. in Egypt) and, presumably, at least a small portion of the events identified in SEVIRI data (Section 2.1) did too. If I'm right about your methodology and you calculated dust mass for all plumes, this means your "Total" mass stats (and statements about this) are missing emission from these events? Even if this is a minor proportion of the total mass emitted from the 16 main source regions, it still seems worthy of mention.

Response: These dust events were attributed to the nearest dust source region based on its upwind edge. This information has been added in section 2.3: 'Once the major dust source regions were established, all dust plumes of the 2015-2017 wintertime dust seasons were assigned to the nearest major source region based on its upwind edge and then assigned the predominant geomorphic class of the region they originated from'

P8, L19: Can you give mean AOD for the other source regions, in addition to the Bodélé Depression? It would be interesting to see whether there are significant differences between sources. For example, some source areas (such as Sudan and Ennedi, both alluvial) seem to be frequently active, but emit a comparatively small amount of dust (and the converse is true for, e.g. Algeria – paleolake). It would be interesting to know if this is a function of differences in mean AOD or plume size. Given your comments about differences in the biogeochemical make up of dust from different source geomorphologies, it seems plausible that this might in some way affect the observed AOD values from specific source types too. Alternatively, it could say something about differences in erosivity/erodibility in these regions. Either way, worthy of comment as this is useful knowledge for remote sensing/dust modelling communities (as well as others!)

Response: Thanks for the excellent suggestion, we have calculated the mean AOD per source region and incorporated this information in Table 2.

P9, L4-5 & Table 2: A reminder (possibly just in Table 2 caption) that South Air and Erg Chech dust mass was apportioned 50/50 to the paleolake and alluvial deposit categories would be useful, as it took me a while to work out how exactly you got to the 64% & 36% numbers.

Response: The reminder has been added.

P9, L11-12: "The ten largest dust storms per season emitted 37.6% of the total dust mass." What proportion of the total detected dust storms did the ten largest dust storms per season account for? This would be useful to know to contextualise the 37.6% figure.

Response: We have added the information per season and revised the sentence to: 'During the 2015-2016 dust season 54.6-84.3 Tg of dust was emitted, of which the ten largest dust storms contributed 33.3%, while the ten largest dust storms of the 2016-2017 dust season produced 45.8% of the 27.7-42.7 Tg emitted that season.'

Technical corrections (typos, citations, etc)

P5, L4: add a reference to Figure 2 (as this visualises pink dust)?

Response: A reference to the Dust RGB methodology? We have now added the reference to Lensky & Rosenfeld, 2008.

P5, L5: ". . .and blue the 10.8 µm channel". I suggest the following addition for clarity (since this is not a BTD): ". . .and blue the brightness temperature (BT) of the 10.8 µm channel".

Response: Thanks for the suggestion, we have changed this.

P5, L8: "[the high temporal resolution of SEVIRI] makes it suitable for observing dust storms". I suggest the following addition: "[the high temporal resolution of SEVIRI] makes it suitable for observing the evolution of dust storms". This is the strength of the higher temporal resolution data; you can still "observe" dust storms (or dust plumes, at least) with lower temporal resolution data

Response: This has been amended

P5, L8-9: ". . .its low spatial resolution means it is less suited to determining the location and nature of the sources". I suggest the following addition: ". . .its low spatial resolution means it is less suited to determining the precise location and nature of the sources". The strength of the higher spatial resolution is the precision it affords, since SEVIRI can give the correct location to a scale of 3 km, no?

Response: This has been amended

P8, L13: "During the 2015-2017 winter dust season between 82.3 - 127.0 Tg of dust was emitted". You should include a reference to Table 2 here.

Response: We have included the reference to Table 2.

P12, L15: "N and P macronutrients". Expand abbreviations/chemical symbols?

Response: This has been corrected.